# SmolVLM: Redefining small and efficient multimodal models

**Andrés Marafioti**[*,h]**, Orr Zohar**[*,s]**, Miquel Farré**[*,h]**,**

Merve Noyan[h], Elie Bakouch[h], Pedro Cuenca[h], Cyril Zakka[h], Loubna Ben Allal[h], Anton Lozhkov[h], Nouamane Tazi[h], Vaibhav Srivastav[h], Joshua Lochner[h], Hugo Larcher[h], Mathieu Morlon[h], Lewis Tunstall[h], Leandro von Werra[h], Thomas Wolf[h]

[h]Hugging Face    [s]Stanford University    [*]Equal contribution

Code: https://github.com/huggingface/smollm

## Abstract

Large Vision-Language Models (VLMs) deliver exceptional performance but require significant computational resources, limiting their deployment on mobile and edge devices. Smaller VLMs typically mirror design choices of larger models, such as extensive image tokenization, leading to inefficient GPU memory usage and constrained practicality for on-device applications.

We introduce **SmolVLM**, a series of compact multimodal models specifically engineered for resource-efficient inference. We systematically explore architectural configurations, tokenization strategies, and data curation optimized for low computational overhead. Through this, we identify key design choices that yield substantial performance gains on both image and video tasks within minimal memory footprints.

Our smallest model, SmolVLM-256M, uses less than 1GB GPU memory during inference and outperforms the 300-times larger Idefics-80B model, despite an 18-month development gap. Our largest model, at 2.2B parameters, rivals state-of-the-art VLMs consuming twice the GPU memory. SmolVLM models extend beyond static images, demonstrating robust video comprehension capabilities.

Our results emphasize that strategic architectural optimizations, aggressive yet efficient tokenization, and carefully curated training data significantly enhance multimodal performance, facilitating practical, energy-efficient deployments at significantly smaller scales.

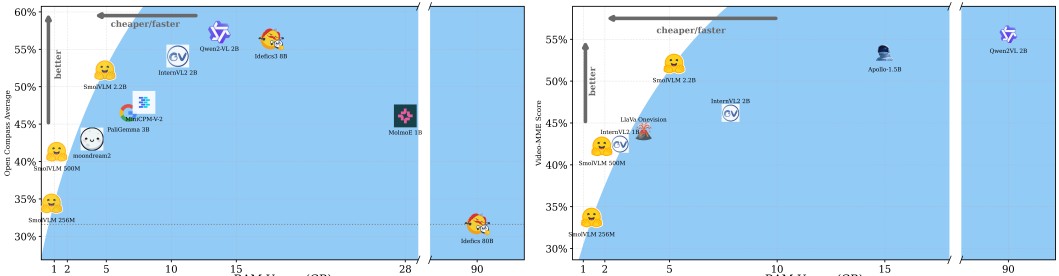

Figure 1: **Smol yet Mighty:** comparison of SmolVLM with other state-of-the-art small VLM models. Image results are sourced from the OpenCompass OpenVLM leaderboard(Duan et al., 2024).

## 1   Introduction

Vision-Language Models (VLMs) have rapidly advanced in capability and adoption (Achiam et al., 2023; Bai et al., 2023; Beyer et al., 2024; Chen et al., 2024c; McKinzie et al., 2024), driving breakthroughs in cross-modal reasoning (Liu et al., 2024a; 2023) and document understanding (Appalaraju et al., 2021; Faysse et al., 2024; Livathinos et al., 2025; Nassar et al., 2025). However, these improvements typically entail large parameter counts and high computational demands.

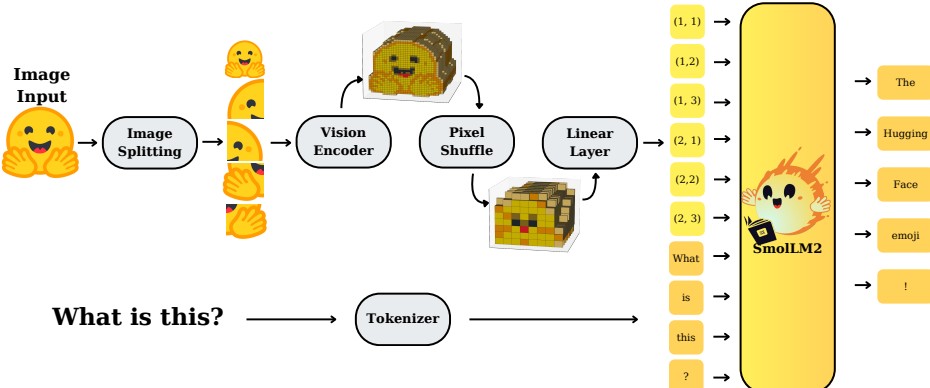

Figure 2: **SmolVLM Architecture.** Images are split into subimages, and frames are sampled from videos, and then encoded into visual features. These features are first rearranged via a pixel-shuffle operation, then mapped into the LLM input space as visual tokens using an MLP projection. Visual tokens are then concatenated/interleaved with text embeddings (orange/red). This combined sequence is passed to the LLM for text output.

Since early large-scale VLMs like Flamingo (Alayrac et al., 2022a) and Idefics (Laurençon et al., 2023) demonstrated capabilities with 80B parameters, new models have slowly appeared at smaller sizes. However, these models often retain high memory demands due to architectural decisions made for their larger counterparts. For instance, Qwen2-VL (Wang et al., 2024) and InternVL 2.5 (Chen et al., 2024b) offer smaller variants (1B-2B), but retain significant computational overhead. Conversely, models from Meta (Llama 3.2 (Dubey et al., 2024)) and Google (Gemma 3) reserve vision capabilities for large-scale models. Even PaliGemma (Beyer et al., 2024), initially efficiency-focused, scaled up significantly in its second release (Steiner et al., 2024). In contrast, Moondream(Korrapati, 2024) keeps focusing on improving performance while maintaining efficiency, and H2OVL-Mississippi(Galib et al., 2024) explicitly targeted on-device deployment. Efficient processing is particularly critical for video understanding tasks, exemplified by Apollo (Zohar et al., 2024b), where memory management is essential. Furthermore, reasoning LLMs generate more tokens during inference, compounding computational costs (DeepSeek-AI, 2025; OpenAI et al., 2024). Therefore, efficiency per token becomes vital to ensure models remain practical for real-world use. *Our contributions are:*

- **Compact yet Powerful Models**: We introduce SmolVLM, a family of powerful small-scale multimodal models, demonstrating that careful architectural design can substantially reduce resource requirements without sacrificing capability.
- **Efficient GPU Memory Usage**: Our smallest model runs inference using less than 1GB GPU RAM, significantly lowering the barrier to on-device deployment.
- **Systematic Architectural Exploration**: We comprehensively investigate the impact of architectural choices, including encoder-LM parameter balance, tokenization methods, positional encoding, and training data composition, identifying critical factors that maximize performance in compact VLMs.
- **Fully Open-source Resources**: To promote reproducibility and facilitate further research, we release all model weights, training datasets, and code, including a mobile application showcasing inference on a smartphone.

## 2   Smoller Model Architecture

### 2.1   How to assign compute between vision and language towers?

VLMs utilize vision encoders (see Figure 2) to generate 'vision tokens' that are then fed into a LM. We investigate optimal capacity allocation between vision encoders and language models (LMs) in compact VLMs. Specifically, we pair three SmolLM2 variants (135M, 360M, and 1.7B parameters) with two SigLIP encoders: a compact 93M SigLIP-B/16 and a larger 428M SigLIP-SO-400M. Typically, larger VLMs disproportionately allocate parameters to the LM; however, as the LM is scaled down, this is no longer the case.

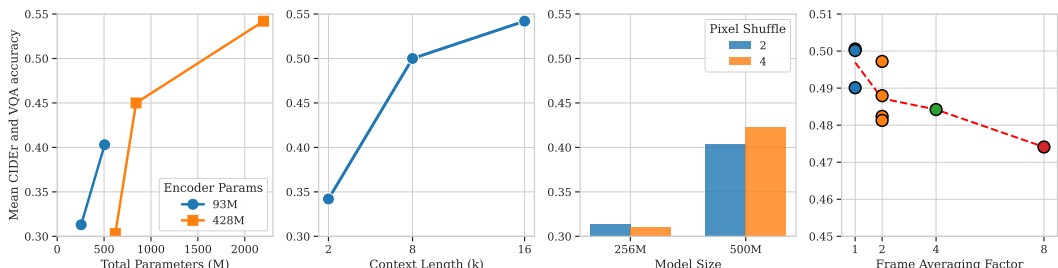

Figure 3: **Performance analysis of SmolVLM configurations.** *(Left)* Impact of vision encoder and language model sizes. Smaller language models (135M) benefit less from larger vision encoders (SigLIP-SO-400M, 428M) compared to SigLIP-B/16 (93M), while larger language models gain more from powerful encoders. *(Middle-left)* Performance significantly improves with increased context lengths (2k to 16k tokens). *(Middle-right)* Optimal pixel shuffle factor (PS=2 vs. PS=4) varies by model size. *(Right)* Frame averaging reduces video performance, with rapid decline as more frames are averaged. Metrics average CIDEr (captioning) and accuracy (visual question answering).

Figure 3 (left) confirms that performance declines significantly when using a large encoder with the smallest LM (135M), highlighting an inefficient encoder-LM balance. At an intermediate LM scale (360M), the larger encoder improves performance by 11.6%, yet this comes with a substantial 66% increase in parameters, making the compact encoder preferable. Only at the largest LM scale (1.7B), the larger encoder represent just a 10% parameter increase.

> *Finding* **1.** Compact multimodal models benefit from a balanced encoder-LM parameter allocation, making smaller vision encoders preferable for efficiency.

## 2.2   How can we efficiently pass the images to the Language Model?

Following Laurençon et al. (2024), we adopt a self-attention architecture in which visual tokens from the vision encoder are concatenated with textual tokens and jointly processed by a language model (e.g., FROMAGe (Koh et al., 2023), BLIP-2 (Li et al., 2023a)). This design requires significantly more context than the 2k-token limit used in SmolLM2, as a single 512 × 512 image encoded with SigLIP-B/16 requires 1024 tokens. To address this, we extended the context capacity by increasing the RoPE base from 10k to 273k, following Liu et al. (2024c), and fine-tuned the model on a mix of long-context data (Dolma books (Soldaini et al., 2024), The Stack (Kocetkov et al., 2022)) and short-context sources (FineWeb-Edu (Penedo et al., 2024), DCLM (Li et al., 2024a), and math from SmolLM2).

While fine-tuning was stable at 16k tokens for the 1.7B LM, smaller models (135M, 360M) struggled beyond 8k. Experiments with our 2.2B SmolVLM confirmed consistent performance gains up to 16k tokens (Figure 3, middle). Accordingly, we adopt a 16k-token context for SmolVLM and an 8k-token limit for smaller variants.

> *Finding* **2.** Compact VLMs significantly benefit from extended context lengths.

Extending the context window alone is not sufficient. Recent VLMs (e.g., MM1 (McKinzie et al., 2024), MiniCPM-V (Yao et al., 2024), InternVL (Chen et al., 2024c)) combine the self-attention architecture with token compression techniques (Zohar et al., 2024b; Laurençon et al., 2024) to fit longer sequences efficiently and reduce computational overhead.

One particularly effective compression method is *pixel shuffle* (space-to-depth), originally proposed for super-resolution tasks (Shi et al., 2016) and recently adopted by Idefics3. Pixel shuffle rearranges spatial features into additional channels, reducing spatial resolution but increasing representational density (Fig. 4). This reduces the total number of visual tokens by a factor of $r^2$, where $r$ is the shuffle ratio. However, higher ratios collapse larger spatial regions into single tokens, impairing tasks requiring precise localization, such as OCR. Models like InternVL and Idefics3 use $r = 2$ to balance compression and spatial fidelity. In contrast, our experiments (Fig. 3, right) show that smaller VLMs benefit from

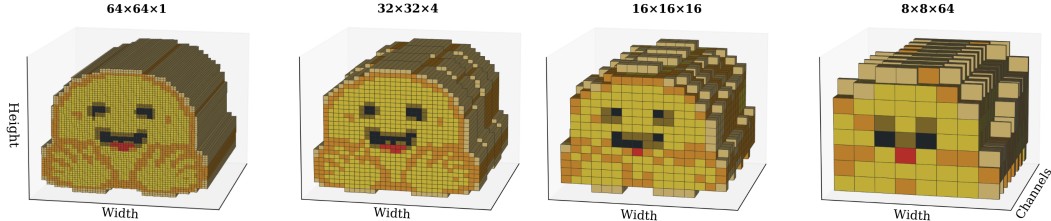

**64×64×1**  **32×32×4**  **16×16×16**  **8×8×64**

Figure 4: **Pixel shuffle.** Rearranges encoded images, trading spatial resolution for increased channel depth. This reduces visual token count while preserving information density.

more aggressive compression ($r = 4$) as the reduced token count eases attention overhead and improves long-context modeling.

> *Finding* **3.** Small VLMs benefit from more aggressive visual token compression.

## 2.3 How can we efficiently encode images and videos?

Balancing token allocation between images and videos is crucial for efficient multimodal modeling: images benefit from higher resolution and more tokens to retain fidelity, whereas videos typically require fewer tokens per frame to handle longer sequences efficiently.

To achieve this, we successfully adopted an image-splitting strategy inspired by UReader (Ye et al., 2023) and SPHINX (Lin et al., 2023b), where high-resolution images are divided into multiple sub-images along with a downsized version of the original. This approach proved effective in maintaining image quality without excessive computational overhead. For videos, however, we found that strategies such as frame averaging, inspired by Liu et al. (2024f), negatively impacted performance. As shown in Figure 3 (right), combining multiple frames significantly degraded OpenCompass-Video results, particularly at higher averaging factors (2, 4, 8). Consequently, frame averaging was excluded from SmolVLM's final design, and videos frames were instead rescaled to the resolution of the image encoder.

> *Finding* **4.** For small models, image splitting enhances performance for vision tasks, whereas video frame averaging does not.

# 3 Smol Instruction Tuning

## 3.1 Learned Tokens vs. String

A major design consideration in SmolVLM involves encoding split subimage positions effectively. Initially, we tried to use simple string tokens (e.g., <row_1_col_2>), which caused early training plateaus - called the "OCR loss plague" - characterized by sudden loss drops without corresponding improvements in OCR performance (Figure 5, left, middle).

To address instability during training, we introduced positional tokens, significantly improving training convergence and reducing stalls. Although larger models were relatively robust to using raw string positions, smaller models benefited substantially from positional tokens, achieving notably higher OCR accuracy and improved generalization across tasks. Figure 5 (center) shows that learned positional tokens consistently outperform naive string positions on multiple image and text benchmarks. Additionally, Figure 5 (right) illustrates that models leveraging learned tokens consistently score higher in both OpenCompass-Image and OpenCompass-Video evaluations, underscoring the effectiveness of structured positional tokenization in compact multimodal models.

> *Finding* **5.** Learned positional tokens outperform raw text tokens for compact VLMs.

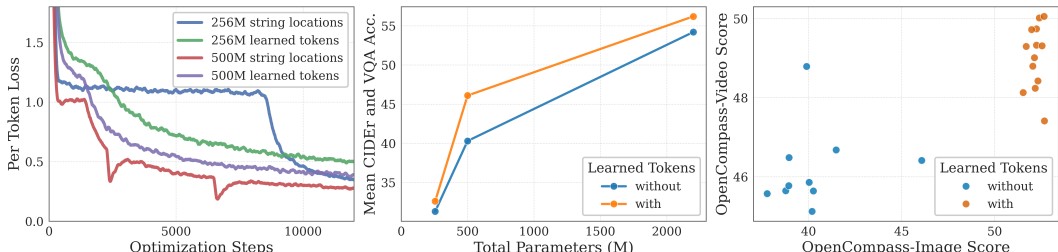

Figure 5: **Tokenization Strategy Comparisons.** *(Left)* Training loss curves illustrating the "OCR loss plague" when using string-based tokens in smaller models. *(Center)* Aggregated evaluation metrics showing consistently higher scores with learned tokens (orange). *(Right)* Scatter plot of OpenCompass-Image vs. OpenCompass-Video: learned tokens dominate the higher-scoring region, especially in image-intensive tasks.

## 3.2 Structured Text Prompts and Media Segmentation

We evaluated how system prompts and explicit media intro/outro prefixes incrementally improve SmolVLM's performance on image (left) and video (right) benchmarks, as shown in Figure 6. Each violin plot represents three checkpoints for a given configuration.

**System Prompts.** We prepend concise instructions to clarify task objectives and reduce ambiguity during zero-shot inference. For example, conversational datasets utilize prompts like *"You are a useful conversational assistant,"* whereas vision-focused tasks employ *"You are a visual agent and should provide concise answers."* The second violin plot in each subplot (Fig. 6) illustrates clear performance improvements from incorporating these system prompts, particularly evident in image-centric tasks.

**Media Intro/Outro Tokens.** To clearly demarcate visual content, we introduce textual markers around image and video segments (e.g., "*Here is an image...*" and "*Here are N frames sampled from a video...*"). The outro tokens then transition back to textual instructions (e.g., "*Given this image/video...*"). The third violin indicates that this strategy substantially boosts performance on video tasks—where confusion between multiple frames is more likely—and still yields measurable improvements on image tasks.

**Masking User Prompts** Drawing on techniques from Allal et al. (2025), we explore user-prompt masking during supervised fine-tuning as a way to reduce overfitting. The right violin plot in Figure 6 shows that masking user queries (orange) yields improved performance in both image and video tasks, compared to the unmasked baseline (blue). This effect is especially pronounced in multimodal QA, where questions are often repetitive and can be trivially memorized by the model. Masking thus forces SmolVLM to rely on task-related content rather than superficial repetition, promoting better generalization.

> *Finding* **6.** System prompts and media intro/outro tokens significantly improve compact VLM performance, particularly for video tasks. During SFT, only train on completions.

## 3.3 Impact of Text Data Reuse from LLM-SFT

A seemingly intuitive practice is to reuse text data from the final supervised fine-tuning stages of large language models, anticipating in-distribution prompts and higher-quality linguistic inputs. However, Figure 7 (left) shows that incorporating LLM-SFT text data (*SmolTalk*) can degrade performance in smaller multimodal architectures—by as much as 3.7% in video tasks and 6.5% in image tasks. We attribute this negative transfer to reduced data diversity, which outweighs any benefits of reusing text. In keeping with Zohar et al. (2024b), we therefore maintain a strict 14% text proportion in our training mix. These findings highlight the importance of a carefully balanced data pipeline, rather than direct adoption of large-scale SFT text for small-scale multimodal models.

> *Finding* **7.** Adding text from SFT blend proved worse than new text SFT data.

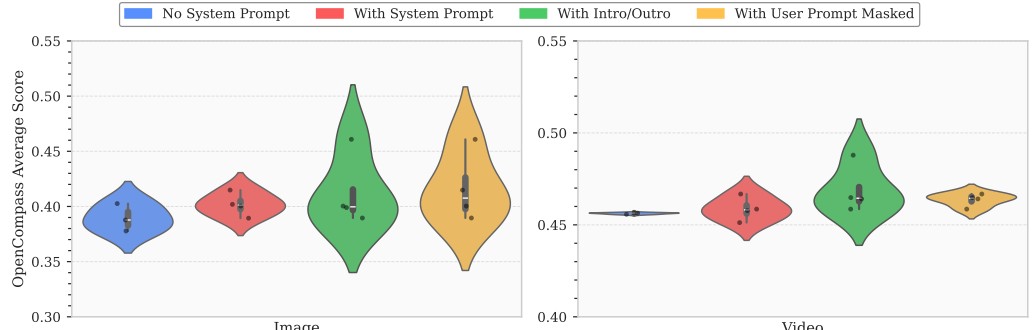

Figure 6: **Cumulative Effect of Training Strategies on SmolVLM Performance.** The visualization shows the progression of performance improvements as different tokenization and prompt engineering strategies are applied sequentially to the SmolVLM base model. *(Left)* Image benchmark results show consistent improvements with each added strategy. *(Right)* Video benchmark results reveal similar patterns with more pronounced gains.

### 3.4 Optimizing Chain-of-Thought Integration for Compact Models

Chain-of-Thought (CoT) prompting, which exposes models to explicit reasoning steps during training, generally enhances reasoning capabilities in large models. However, its effect on smaller multimodal architectures remains unclear. To investigate this, we varied the proportion of CoT data integrated into the Mammoth dataset (Yue et al., 2024b), covering text, image, and video tasks. Figure 7 (middle) shows that incorporating a minimal fraction (0.02–0.05%) of CoT examples slightly improved performance, but higher proportions markedly degraded results—especially in image tasks. These observations suggest that excessive reasoning-oriented textual data can overwhelm the limited capacity of smaller VLMs, thereby compromising their visual representation capabilities. Consequently, compact models benefit most from very sparse inclusion of CoT data rather than the extensive use typically beneficial in larger-scale architectures.

> *Finding* 8. Excessive CoT data harms compact model performance.

### 3.5 Impact of Video Sequence Length on Model Performance

Increasing video duration during training offers richer temporal context but comes at greater computational cost. To identify an optimal duration, we trained SmolVLM on average video lengths ranging from 1.5 to 3.5 minutes. Figure 7 (right) demonstrates clear performance improvements for both video and image benchmarks as video durations approached approximately 3.5 minutes, likely due to more effective cross-modal feature learning. Extending video duration beyond 3.5 minutes yielded minimal further gains, indicating diminishing returns relative to the added computational expense. Thus, moderately extending video sequences enhances performance significantly in smaller models, whereas overly long sequences do not proportionally justify their computational cost.

> *Finding* 9. Moderately increasing video duration during training improves both video and image task performance in compact VLMs.

## 4 Experimental Results

We construct three variants of SmolVLM, tailored to different computational environments:

- **SmolVLM-256M**: Our smallest model, combining the 93M-parameter SigLIP-B/16 (Zhai et al., 2023) and the SmolLM2-135M (Allal et al., 2025). It operates on < 1GB GRAM, making it ideal for resource-constrained edge application.

- **SmolVLM-500M**: A mid-range model featuring the same 93M-parameter SigLIP-B/16 paired with the larger SmolLM2-360M. It balances memory efficiency and performance and is suitable for moderate-resource edge devices.

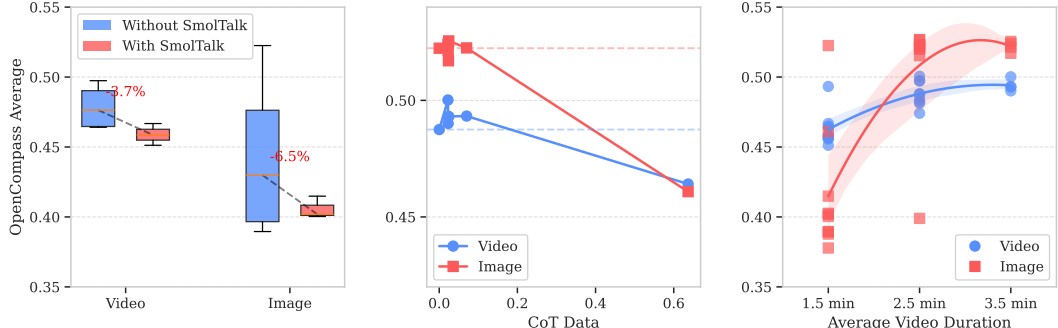

Figure 7: **Impact of Training Strategies on Smol-Scale Multimodal Models.** *(Left)* Reusing text data from LLM-SFT (*SmolTalk*) reduces both image and video scores in smaller models. *(Middle)* A minimal fraction (0.02%–0.05%) of Chain-of-Thought (CoT) data yields optimal results, while heavier CoT usage degrades performance. *(Right)* Increasing average video duration beyond 3.5 min leads to diminished returns for both image and video tasks.

- **SmolVLM-2.2B**: The largest variant, with a 400M-parameter SigLIP-SO400M and a 1.7B-parameter SmolLM2 backbone. This model maximizes performance while remaining deployable on higher-end edge systems.

## 4.1 Training Data

Model training proceeds in two stages: (1) a vision stage, and (2) a video stage. The vision training stage uses a new mixture of the datasets used in Laurençon et al. (2024), to which we added MathWriting (Gervais et al., 2024). The mixture was balanced to emphasize visual and structured data interpretation while maintaining the focus on reasoning and problem-solving capabilities. The visual components comprise document understanding, captioning and visual question answering (including 2% dedicated to multi-image reasoning), chart understanding, table understanding, and visual reasoning tasks. To preserve the model's performance in text-based tasks, we retained a modest of general knowledge Q&A and text-based reasoning & logic problems, which incorporate mathematics and coding challenges.

The video fine-tuning stage maintains 14% of text data and 33% of video to achieve optimal performance, following the learnings of Zohar et al. (2024b). For video, we sample visual description and captioning from LLaVA-video-178k (Zhang et al., 2024), Video-STAR (Zohar et al., 2024a), Vript (Yang et al., 2024), and ShareGPT4Video (Chen et al., 2023), temporal understanding from Vista-400k (Ren et al., 2024), and narrative comprehension from MovieChat (Song et al., 2024) and FineVideo (Farré et al., 2024). Multi-image data was sampled from M4-Instruct (Liu et al., 2024a) and Mammoth (Guo et al., 2024). The text samples were sourced from (Xu et al., 2024).

For a more granular description, Figure 8 provides a detailed overview of the training data distribution used in both our vision and video fine-tuning stages.

## 4.2 Evaluation details

We evaluated SmolVLM using VLMEvalKit (Duan et al., 2024) to ensure reproducibility. The full results are available online[1]. Currently, OpenVLM Leaderboard covers 239 different VLMs and 31 different multi-modal benchmarks. Further, we plot the performance against the RAM required to run the evaluations. We argue that model size is usually used as a proxy for the computational cost required to run a model. This is misleading for VLMs because the architecture strongly influences how expensive it is to run the model, in our opinion RAM usage is a better proxy. For the experiments, we use each model's default input image resizing. For SmolVLM, this resizes the longest edge of images to 1536 in the 256M and 500M models and 1920 in the 2.2B one.

---

[1]OpenVLM Leaderboard

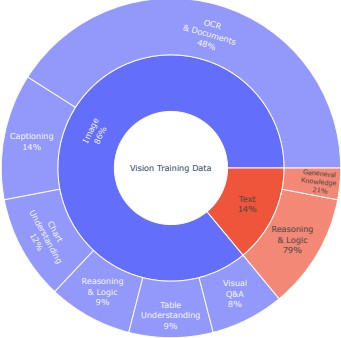 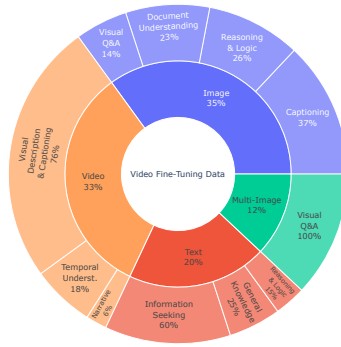

Figure 8: **Data Details.** Training dataset details for Vision *(Left)* and video *(Right)*, broken down by modality and sub-categories.

## 4.3 Strong Performance at a Tiny Scale

We evaluate SmolVLM's performance relative to model size, comparing three variants (256M, 500M, and 2.2B) against an efficient state-of-the-art (SOTA) open-source (OS) model. Table 1 summarizes results across nine demanding vision-language benchmarks and five video benchmarks. We highlight in the table MolmoE 7B with 1B activate parameters (Deitke et al., 2024)(MolmoE-A1B-7B) for vision tasks and InternVL2-2B Chen et al. (2024c) for video tasks. A broader array of competing models are showcased in Fig. 1.

**Efficiency and Memory Footprint.**    SmolVLM demonstrates remarkable computational efficiency compared to significantly larger models. Single-image inference requires only 0.8GB of VRAM for the 256M variant, 1.2GB for the 500M, and 4.9GB for the 2.2B—dramatically lower than 27.7GB required by MolmoE-A1B-7B. Even compared to models of similar parameter scales, SmolVLM is notably more efficient: Qwen2VL-2B requires 13.7GB VRAM and InternVL2-2B requires 10.5GB VRAM, highlighting that parameter count alone does not dictate compute requirements. At batch size 64, memory usage for SmolVLM remains practical: 15.0GB (256M), 16.0GB (500M), and 49.9GB (2.2B). These results highlight SmolVLM's substantial advantages for deployment in GPU-constrained environments.

**Overall Gains from Scaling.**    Increasing SmolVLM's parameter count consistently yields substantial performance improvements across all evaluated benchmarks. The largest model (2.2B) achieves the highest overall score at 59.8%, followed by the intermediate 500M variant (51.0%) and the smallest 256M variant (44.0%). Notably, even the smallest SmolVLM-256M significantly surpasses the much larger Idefics 80B model (see Fig. 1) on nearly all benchmarks, emphasizing effective vision capabilities at modest scales. The few exceptions—particularly MMMU (29.0% vs. 42.3%) and AI2D (46.4% vs. 56.3%)—highlight benchmarks where strong linguistic reasoning from a large language backbone remains crucial. Intriguingly, visually oriented tasks such as OCRBench also benefit markedly from scaling language model capacity, with a nearly 10-point improvement when moving from 256M (52.6%) to 500M (61.0%). These results underscore that larger language models provide enhanced context management and improved multimodal reasoning, benefiting both language-intensive and vision-centric tasks.

**Comparison with Other Compact VLMs.**    Figure 1 situates SmolVLM-2.2B among recent small-scale VLMs by comparing OpenCompass benchmark performance against GPU memory consumption per image. SmolVLM-2.2B achieves notably strong performance on MathVista (51.5) and ScienceQA (90.0), while maintaining exceptionally low GPU usage of just 4.9GB VRAM. In contrast, models requiring significant more compute offer such as Qwen2VL-2B and InternVL2-2B aren't clearly better performers. Specifically, Qwen2VL-2B slightly surpasses SmolVLM-2.2B on AI2D (74.7 vs. 70.0) and ChartQA (73.5 vs. 68.8), yet falls short on MathVista (48.0 vs. 51.5) and ScienceQA (78.7 vs. 90.0). Similarly, InternVL2-2B

| Capability | Benchmark | SmolVLM 256M | SmolVLM 500M | SmolVLM 2.2B | Efficient OS |
|---|---|---|---|---|---|
| Single-Image | OCRBench (Liu et al., 2024e) *Character Recognition* | 52.6% | 61.0% | 72.9% | 54.7% *MolmoE-A1B-7B* |
| | AI2D (Kembhavi et al., 2016) *Science Diagrams* | 46.4% | 59.2% | 70.0% | 71.0% *MolmoE-A1B-7B* |
| | ChartQA (Masry et al., 2022) *Chart Understanding* | 55.6% | 62.8% | 68.7% | 48.0% *MolmoE-A1B-7B* |
| | TextVQA (Singh et al., 2019) *Text Understanding* | 50.2% | 60.2% | 73.0% | 61.5% *MolmoE-A1B-7B* |
| | DocVQA (Mathew et al., 2021) *Document Understanding* | 58.3% | 70.5% | 80.0% | 77.7% *MolmoE-A1B-7B* |
| | ScienceQA (Lu et al., 2022) *High-school Science* | 73.8% | 80.0% | 89.6% | 87.5% *MolmoE-A1B-7B* |
| Multi-task | MMMU (Yue et al., 2024a) *College-level Multidiscipline* | 29.0% | 33.7% | 42.0% | 33.9% *MolmoE-A1B-7B* |
| | MathVista (Lu et al., 2024b) *General Math Understanding* | 35.9% | 40.1% | 51.5% | 37.6% *MolmoE-A1B-7B* |
| | MMStar (Chen et al., 2024a) *Multidisciplinary Reasoning* | 34.6% | 38.3% | 46.0% | 43.1% *MolmoE-A1B-7B* |
| Video | Video-MME (Fu et al., 2024) *General Video Understanding* | 33.7% | 42.2% | 52.1% | 45.0% *InternVL2-2B* |
| | MLVU (Zhou et al., 2024) *MovieQA + MSRVTT-Cap* | 40.6% | 47.3% | 55.2% | 48.2% *InternVL2-2B* |
| | MVBench (Li et al., 2024b) *Multiview Reasoning* | 32.7% | 39.7% | 46.3% | 60.2% *InternVL2-2B* |
| | WorldSense (Hong et al., 2025) *Temporal + Physics* | 29.7% | 30.6% | 36.2% | 32.4% *Qwen2VL-7B* |
| | TempCompass (Liu et al., 2024d) *Temporal Understanding* | 43.1% | 49.0% | 53.7% | 53.4% *InternVL2-2B* |
| **Overall Avg.** | Across Benchmarks | 44.0% | 51.0% | 59.8% | – |
| RAM Usage | Batch size = 1 | 0.8 GB | 1.2 GB | 4.9 GB | 27.7 GB *MolmoE-A1B-7B* |
| | batch size = 64 | 15.0 GB | 16.0 GB | 49.9 GB | – |

Table 1: Performance comparison between SmolVLM models (256M–2.2B) and efficient open source models on vision- and video-language benchmarks. RAM is reported in GB.

achieves higher scores on ScienceQA (94.1 vs. 90.0) and MMStar (49.8 vs. 46.0), but at more than double the VRAM cost.

Further comparisons highlight distinct trade-offs among size, memory footprint, and task-specific performance. MiniCPM-V2 (2.8B parameters) underperforms SmolVLM-2.2B on most benchmarks. Other models such as Moondream2 and PaliGemma (both around 2–3B parameters) exhibit significant variance across tasks: Moondream2, for instance, scores well on ChartQA (72.2) with just 3.9GB VRAM but substantially underperforms on MMMU (29.3). Conversely, PaliGemma excels at ScienceQA (94.3) yet struggles on ChartQA (33.7). This variability underscores how specialized training impacts per-task.

**Video Benchmarks.** Table 1 provides comprehensive results across five diverse video benchmarks: Video-MME, MLVU, MVBench, TempCompass, and WorldSense. SmolVLM-2.2B notably excels at Video-MME (52.1) and WorldSense (36.2), outperforming significantly larger models such as Qwen2 VL-7B (32.4 on WorldSense), showcasing strong capabilities in complex multimodal video comprehension tasks. The SmolVLM-500M variant also demonstrates robust performance, achieving competitive scores on TempCompass (49.0) and WorldSense (30.6), highlighting sophisticated temporal reasoning and real-world visual understanding at a scale ideal for edge-device deployment. Despite their compact parameter counts, SmolVLM variants consistently balance efficient resource use with impressive accuracy, reinforcing their suitability for resource-constrained multimodal scenarios.

### 4.4 On-Device Performance

To comprehensively assess the deployment practicality of SmolVLM, we benchmarked its throughput across varying batch sizes on two representative hardware platforms: NVIDIA A100 and NVIDIA L4 GPUs (see Figure 9). Our evaluations highlight SmolVLM's suitability for on-device and edge deployment scenarios.

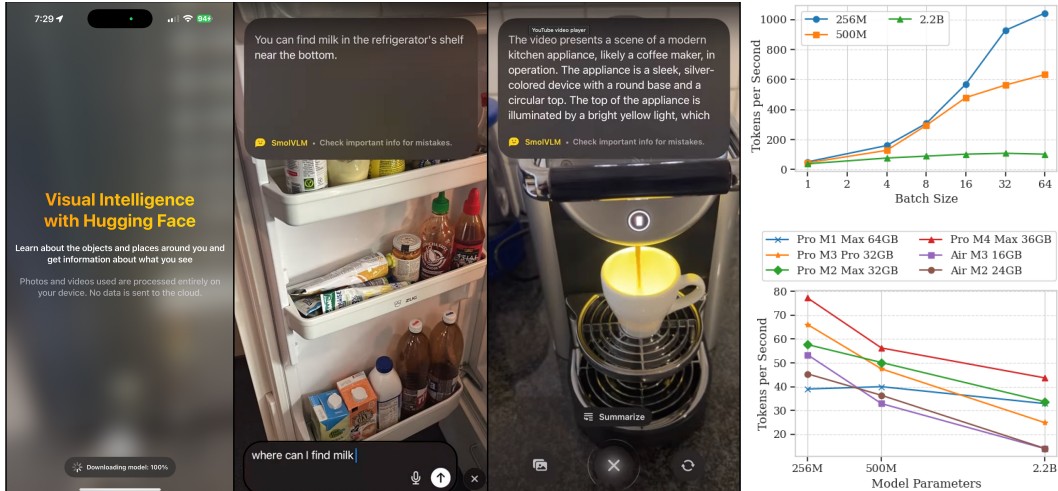

Figure 9: **SmolVLM on edge device.** *(Left)* Examples of the `HuggingSnap` app, where SmolVLM can run locally, on the device, on consumer phones. For example, interactions can be done using a mobile interface to detect objects and answer questions. *(Right)* Throughput in tokens per second on NVIDIA A100 GPUs *(top)* and different consumer personal computers *(bottom)* across different batch sizes and model variants.

On the A100 GPU, the smallest SmolVLM-256M variant achieves impressive throughput, scaling from 0.8 examples per second at batch size 1 to 16.3 examples per second at batch size 64. The 500M variant similarly scales from 0.7 to 9.9 examples per second, while the largest 2.2B variant demonstrates more modest scaling (0.6 to 1.7 examples per second), indicative of its higher computational demands.

Evaluations on the L4 GPU further emphasize SmolVLM's edge compatibility. Here, the 256M variant reaches peak throughput at 2.7 examples per second with batch size 8, subsequently diminishing due to memory constraints. The 500M and 2.2B variants peak at lower batch sizes (1.4 and 0.25 examples per second, respectively), underscoring their efficiency even under more restrictive hardware conditions.

Finally, we accompany the release with several optimized ONNX (Open Neural Network Exchange) exports, facilitating cross-platform compatibility and broadening deployment opportunities across consumer-grade hardware targets. Notably, we demonstrate the ability to efficiently run these models locally within a browser environment via WebGPU, with the 256M variant achieving up to 80 decode tokens per second on a MacBook Pro (M4 Max).

## 5   Conclusion

We introduced **SmolVLM**, a family of memory-efficient Vision-Language Models ranging from 256M to 2.2B parameters. Remarkably, even our smallest variant requires less than 1GB of GPU memory yet surpasses state-of-the-art 80B-parameter models from just 18 months ago (Laurençon et al., 2023). Our findings emphasize a critical insight: scaling down large VLM architectures optimized under resource-rich conditions results in disproportionately high memory demands during inference with little advantage over specialized architectures. By contrast, SmolVLM's design philosophy explicitly prioritizes compact architectural innovations, aggressive but careful tokenization methods, and efficient training strategies, enabling powerful multimodal capabilities at a fraction of the computational cost.

All model weights, training datasets, and training code are publicly released to encourage reproducibility, transparency, and continued innovation. We hope SmolVLM will inspire the next generation of lightweight, efficient VLMs, unlocking new possibilities for real-time multimodal inference with minimal power consumption.

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
