# OpenReview forum: "SmolVLM: Redefining small and efficient multimodal models"
_colmweb.org/COLM/2025/Conference — COLM 2025_

### Official Review · Reviewer_BnL7 · 2025-04-15

**Rating:** 8
**Confidence:** 4
**Ethics Flag:** 1

**Summary:**

The paper presents SmolVLM, a series of carefully engineered small-scale multimodal models. It provides a detailed comparison of different model architectures, tokenization strategies, and data curation methods, analyzing their impact on overall performance. A set of model weights is released, achieving strong performance gains with significantly lower memory consumption.

**Questions To Authors:**

1. In Section 3.1, the term positional tokens is mentioned. Could the authors clarify what this refers to, and how it relates to position embeddings? An example would be helpful to illustrate the distinction.

**Reasons To Accept:**

1. The paper is well-structured and presents its findings clearly.
2. The experimental results are solid and provide valuable engineering insights for the community.

**Reasons To Reject:**

1. In Section 2.1, the paper compares the impact of vision encoders with different parameter scales on model performance. However, it is unclear whether the input image resolutions are kept consistent across these encoders. How does the paper isolate the effect of encoder size from that of image resolution?
2. In Figure 3 (left), the authors state that smaller language models (LMs) benefit less from larger vision encoders. However, since each curve represents a fixed encoder, using curves corresponding to fixed LMs instead might improve interpretability.
3. In Figure 3 (right), the legend is missing—it's unclear what the different colors represent.
4. Lines 98–99 claim that smaller VLMs benefit from more aggressive compression, but Figure 3 (middle-right) seems to show the opposite: the 256M model's performance decreases while the 500M model's performance improves.
5. Line 186 states that training beyond 3.5 minutes yields minimal further gains, yet Figure 7 (right) does not show any data points beyond the 3.5-minute mark.
6. In addition to the composition of the training data, more details about the training hyperparameters would be helpful—for example, which modules were frozen or trained during different training stages.

---

> ### Author Response · Authors · 2025-06-01
> **Responce to R-BnL7**
>
> We sincerely thank reviewer R-BnL7 for their detailed and comprehensive feedback. We are pleased with the thorough review full of actionable points. We are happy that the reviewer finds our insights useful for the community.
>
> 1. While we could not control for encoder resolution, with image splitting, all experiments had comparable
> Indeed, vision encoders with larger input resolutions have greater capacity, but at that time, there wasn't a Siglip So 400M with an input resolution larger than 384, so we couldn't make a comparison with the same input resolution. We also didn't want to limit our small vision encoder by using one with a lower resolution. However, since we are employing image splitting, the effective overall image resolution during training is comparable. During evaluation, images are rarely small enough that they wouldn't be split, so both models handle them at similar resolutions.
> 2. We will modify Figure 3 (left) to use curves corresponding to fixed LM sizes for readability. Thank you for the suggestion.
> 3. We will improve Figure 3 (right) for readability. The different colors are not necessary – there is no difference.
> 4. In Figure 3, the performance difference between the 256M model at r=2 and r=4 falls within the noise of the benchmarks, making it difficult to determine which is superior. However, for the 500M model, it is evident that r=4 performs better. Notably, r=4 significantly reduces the number of tokens required for both models, leading to a much more efficient model. Therefore, we assert that r=4 is clearly beneficial for both model sizes. We will clarify that when we say "small models benefit from r=4," we mean that although the quality may not necessarily improve, the quality/performance trade-off is quite favorable.
> 5. This is because the difference between the 1.5 and 2.5 average durations was much larger than the difference between the 2.5 and 3.5 average durations. We will clarify that the intention is to start saturating at 3 minutes, not 3.5 minutes.
> 6. We will add the following table to our manuscript:
>
>
> | Category | Setting | Alignment | Pre-training | SFT (Single-Image) | SFT (Image + Video) |
> |----------|---------|-----------|--------------|--------------------|---------------------|
> | **Vision** | Resolution | 512 / 384 | 512 / 384 × {1×1 … 6×6} | 512 / 384 × {1×1 … 6×6} | 512 / 384 × {1×1 … 6×6} |
> |          | # Tokens | 64 / 81 | max 64 / 81 × 17 | max 81 × 17 | max 64 / 81 × 17 |
> | **Data** | Dataset | Images | High-quality Images | Instructional Images | Multi-Image & Video |
> |          | # Samples | 4 M | 20 M | 13.5 M | 3 M |
> | **Model** | Trainable params | MLP + LoRA (all) | Full model | Full model | Full model |
> |          | 256 M LLM | ≈ 0.9 M | 256 M | 256 M | 256 M |
> |          | 500 M LLM | ≈ 1.8 M | 500 M | 500 M | 500 M |
> |          | 2.2 B LLM | ≈ 7.5 M | 2.2 B | 2.2 B | 2.2 B |
> | **Training** | Batch size | 1024 | 2048 | 8192 | 512 |
> |          | LR (ψ_vision) | 1 × 10^-4 | 1 × 10^-5 | 1 × 10^-5 | 5 × 10^-6 |
> |          | LR (θ_proj)   | 1 × 10^-3 | 1 × 10^-5 | 1 × 10^-5 | 1 × 10^-4 |
> |          | LR (φ_LLM)    | 1 × 10^-3 | 1 × 10^-5 | 1 × 10^-5 | 2 × 10^-5 |
> |          | Epochs | 1 | 1 | 1 | 1 |
>
> Along with an experimental section detailing additional training details.
>
> Answer to Questions:
> Term positional tokens: When we split images to pass them to the LLM, we added special tokens to the text tokenizer, which are later learned during SFT. This is the effect of concatenating a learnable embedding before each image patch, allowing the model to more easily identify it. Thank you for pointing out that this wasn't very clear in the original manuscript; we will revise it to enhance clarity.

---

### Official Review · Reviewer_jgcB · 2025-05-13

**Rating:** 7
**Confidence:** 4
**Ethics Flag:** 1

**Summary:**

The paper describes a series of compact multimodal models specifically engineered for resource-efficient inference. First the authors identify key design criteria to be optimized for low computational overhead—architectural configurations, efficient tokenization strategies, and training data curation. Then they validate the design choices by showing how the resulting vision-language models (VLM) show substantial performance gains on both image and video tasks with minimal memory footprints, making them practically viable for multimodal tasks.

**Reasons To Accept:**

1. Well-organized, well-structured and clearly written paper.
2. The paper lays out a set of design choices to reduce the size of multimodal models. Experiments validate these design choices in multimodal settings.
3. The datasets on which the models are validated are diverse (single-image, multitask, and video) and challenging, lending credibility to the experiments and the significance of the contribution.
4. The authors make all model weights, datasets, and code, publicly available. Additionally, they include a mobile application to demonstrate inference on a smartphone with the reduced and compact multimodal model.

**Reasons To Reject:**

1. Evaluation metric insufficiency: The authors evaluate the resulting efficiency of the reduced model size in terms of the amount of RAM required to run the evaluations. Although the architecture is an important determinant of the computational cost, it appears that they are missing out on other important aspects of efficiency to convincingly argue about the improved efficiency. Why did the authors not consider aspects related to latency, throughput, and energy consumption? Without this, it is difficult to assess the true benefits of the proposed model reduction for on-device applications. Edit: the author response includes throughput evaluations.

2. A comparison with other on-device multimodal models is also missing, which fails to give a complete picture of the benefits of the proposed reduced model.

---

> ### Author Response · Authors · 2025-06-01
> **Response to R-jgcB**
>
> We sincerely thank Reviewer jgcB for their evaluation and valuable feedback. We are pleased the reviewer recognizes the high bar we set with the evaluations we chose, the thorough layout we set for design choices in reducing the size of multimodal models, and our commitment to open-source contributions, including the mobile application.
>
> Below we address the key points raised:
>
> 1. Evaluation metric insufficiency:
>
> We agree that VRAM alone does not fully validate real-time performance. Thus, we will add additional figures with comprehensive wall-clock throughput metrics (Tokens Per Second, Time-To-First-Token) measured on both consumer hardware (MacBook M1-M4 Max series) and server-side GPUs (H100s). For example, SmolVLM-256M achieves up to 45 tokens/sec on a three-year-old MacBook Air M2 with 16GB of RAM, demonstrating practical real-time capabilities. *More information on this can be found in the general rebuttal.*
>
> 2. Small VLM direct comparison:
>
> Regarding energy efficiency, direct comparisons to other models across diverse hardware remain challenging; however, our results on the H100 GPUs indicate SmolVLM variants consistently outperform comparable models in tokens processed per second by at least 3 times, strongly suggesting superior energy efficiency per token.

---

> > ### Comment · Reviewer_jgcB · 2025-06-10
> > **Adequate response from the authors**
> >
> > I have noted the response from the authors and will change my rating to reflect this.

---

### Official Review · Reviewer_RaaY · 2025-05-13

**Rating:** 5
**Confidence:** 3
**Ethics Flag:** 1

**Summary:**

This paper introduces **SmolVLM**, a family of compact multimodal vision-language models optimized for efficient inference on resource-constrained devices. Through systematic exploration of architecture, tokenization, and training data, the authors show that smaller vision encoders, extended context windows, aggressive visual token compression, and structured positional tokenization significantly enhance performance. The smallest model, SmolVLM-256M, achieves better results than the much larger Idefics-80B model using less than 1GB GPU memory. The largest variant (2.2B parameters) rivals state-of-the-art multimodal models while requiring half the GPU memory. Additionally, SmolVLM demonstrates robust performance on both image and video benchmarks, underscoring its practical applicability. The study emphasizes that careful architectural design and efficient training strategies are crucial for developing small-scale yet powerful multimodal models. All resources, including model weights, training data, and code, are publicly released to promote reproducibility and further innovation.

**Reasons To Accept:**

1. **Innovative and Efficient Model Design**:
   The paper introduces SmolVLM, a compact yet powerful multimodal architecture that outperforms significantly larger models using dramatically less GPU memory, highlighting important advancements in efficient multimodal modeling.

2. **Comprehensive Experimental Validation**:
   Extensive experiments systematically explore architectural choices, tokenization methods, and data strategies, providing practical guidelines and insights valuable for researchers and practitioners interested in deploying vision-language models on resource-constrained edge devices.

3. **Strong Open-Source Contribution**:
   All model weights, training datasets, and source code are publicly available, fostering reproducibility and supporting future research. This openness significantly benefits the research community by providing ready-to-use resources for efficient multimodal model development.

**Reasons To Reject:**

1. **Questionable comparison target in the abstract**
   The abstract highlights a performance comparison against Idefics-80B, which feels out of place because many recent *small-scale* multimodal models would be more relevant baselines.

2. **Primarily engineering-oriented, with limited novelty**
   Most techniques and findings—such as extending context length by adjusting the RoPE base and compressing visual tokens via pixel shuffle—have already been explored in prior work, making the overall contribution incremental rather than groundbreaking.

3. **Incomplete and potentially biased evaluation**
   The experimental section omits direct, side-by-side comparisons with several SOTA *small* VLMs (e.g., MiniCPM-V, NVILA). In addition, the study focuses on leaderboard averages without reporting real-world latency, energy, or on-device throughput—metrics that are critical for a paper whose primary claim is efficiency. This limited evaluation makes it hard to judge whether SmolVLM truly advances the state of the art beyond existing compact models.

---

> ### Author Response · Authors · 2025-06-01
> **Response to R-RaaY**
>
> We thank R-RaaY for their detailed and comprehensive review. We are pleased the reviewer recognizes the significance of targeting edge device limitations, appreciates our systematic design-space exploration, comprehensive benchmarking, and our commitment to open-source contributions, including the mobile application.
>
> Below we address the key points raised:
>
> 1. Questionable comparison target in the abstract
>
> We only compare to Idefics-80B to motivate the exploration of small VLMs, and illustrate how much the field has progressed, as a 256M model outperforms an 80B model from 20 months ago.
> We will revise the abstract and include comparisons to other small, newer VLM models (e.g., InternVL2-1b/LLaVA-OV-0.8B).
> Figure 1 (right below the abstract) contains more relevant comparisons.
>
> 2. Primarily engineering-oriented, with limited novelty
>
> Please note that SmolVLM is the first VLM with fewer than 800M parameters (256M and 500M). This small size allows it to perform inference on devices with very limited resources, and it also enables exceptionally high throughput at large batch sizes.
>
> 3. Incomplete and potentially biased evaluation.
>
> (A) Edge benchmarking
>
> We agree that VRAM alone does not fully validate real-time performance. Thus, we will add comprehensive wall-clock throughput metrics (Tokens Per Second, Time-To-First-Token) measured on both consumer hardware (MacBook M1-M4 Max series) and huge GPUs (H100s). For example, SmolVLM-256M achieves up to 45 tokens/sec on a three-year-old MacBook Air M2 with 16GB of RAM, demonstrating practical real-time capabilities. *More information on this can be found in the general rebuttal.*
>
> (B) Comparisons to other models in the experimental section.
>
> On the paper's first page, we compared SmolVLM to several small VLMs based on average performances. Due to the page limit and the number of experiments shown in the manuscript, we only compared selected models for individual benchmarks. However, to enhance reader convenience, we will include the comprehensive table below, including MiniCPM-V and NVILA.
>
> ### Image-only benchmarks
>
> | Model | Params (B) | OCRBench | AI2D | ChartQA | TextVQA | DocVQA | ScienceQA | MMMU | MathVista | MMStar |
> |-------|:---------:|:-------:|:----:|:------:|:------:|:-----:|:---------:|:----:|:--------:|:-----:|
> | SmolVLM 256 M            | 0.26 | 52.6 | 46.4 | 55.6 | 50.2 | 58.3 | 73.8 | 29.0 | 35.9 | 34.6 |
> | SmolVLM 500 M            | 0.50 | 61.0 | 59.2 | 62.8 | 60.2 | 70.5 | 80.0 | 33.7 | 40.1 | 38.3 |
> | LLaVA-OneVision    | 0.80 | 58.3 | 59.4 | 61.4   | —   | 70.0   | 67.5 | 32.7 | 35.9 | 37.7 |
> | InternVL2-1B             | 0.90 | 75.4 | 64.1 | 72.9 | 70.5 | 81.7 | —   | 36.7 | 37.7 | 45.6 |
> | Moondream 2              | 1.86 | 58.5 | 58.8 | —   | —   | —   | —   | 29.3 | 33.8 | 42.1 |
> | Qwen2-VL-2B              | 2.00 | 79.7 | 74.7 | 73.5 | —   | 90.1 | 78.7 | 42.2 | 48.0 | 47.5 |
> | SmolVLM 2.2 B            | 2.20 | 72.9 | 70.0 | 68.7 | 73.0 | 80.0 | 89.6 | 42.0 | 51.5 | 46.0 |
> | InternVL2-2B             | 2.20 | 78.1 | 74.1 | 71.7 | 73.4 | 86.9 | 94.1 | 36.3 | 47.0 | 49.8 |
> | MiniCPM-V                | 2.80 | 36.6 | —   | —   | 60.6 | 38.2 | —   | 38.3 | 28.9 | —   |
> | MiniCPM-V 2.0            | 2.80 | 60.5 | —   | —   | 74.1 | 71.9 | —   | 38.2 | 38.7 | —   |
> | NVILA-Lite               | 8.00 | —   | 91.0 | 84.8 | 78.1 | 91.7 | —   | 50.7 | 64.5 | —   |
>
>
> ### Video benchmarks
>
> | Model | Params (B) | Video-MME | MLVU | MVBench | WorldSense | TempCompass |
> |-------|:---------:|:---------:|:----:|:-------:|:-----------:|:-----------:|
> | SmolVLM 256 M            | 0.26 | 33.7 | 40.6 | 32.7 | 29.7 | 43.1 |
> | LLaVA-OneVision-0.5 B    | 0.50 | 44.0   | 50.3   | —   | —   | 53.2   |
> | SmolVLM 500 M            | 0.50 | 42.2 | 47.3 | 39.7 | 30.6 | 49.0 |
> | InternVL2-1B             | 0.90 | 42.6 | —   | 57.5 | —   | —   |
> | Qwen2-VL-2B              | 2.00 | 55.6 | —   | 63.2 | —   | —   |
> | SmolVLM 2.2 B            | 2.20 | 52.1 | 55.2 | 46.3 | 36.2 | 53.7 |
> | InternVL2-2B             | 2.20 | 45.0 | 48.2 | 60.2 | —   | 53.4 |
> | NVILA               | 8.00 | 64.2   | 70.1   | 68.1   | —   | —   |

---

### Official Review · Reviewer_ofar · 2025-05-20

**Rating:** 7
**Confidence:** 5
**Ethics Flag:** 1

**Summary:**

This paper introduces SmolVLM, which confronts high memory and compute costs of vision‑language models by pairing compact SigLIP encoders with lightweight SmolLMs, pixel‑shuffle compression, image‑splitting, and extended 8–16k contexts. Three variants (256 M–2.2 B) beat baselines on 14 image and 5 video benchmarks using under 1 GB GPU memory for real‑time edge deployment scenarios.

**Questions To Authors:**

After reading the rebuttal and comments from other reviewers, I adjust the score to 7 due to the feedback on fairness. But still think this work will be a good experimental practical～

**Reasons To Accept:**

1. The work squarely targets the growing gap between state‑of‑the‑art VLM accuracy and the memory/power budgets of edge devices, which is very important for the VLM community.

2. The authors vary encoder/LM size, context length, visual‑token compression, and video sampling, turning the paper into a design‑space exploration rather than a one‑off model proposal. The nine “Findings” distilled from Figure 3 and Section 2 provide reusable insights (e.g., balanced encoder‑LM sizing, pixel‑shuffle ratio, minimal CoT) rather than ad‑hoc tweaks.

3. Three SmolVLM variants (256 M‑2.2 B) beat or match larger open‑source baselines on nine image and five video benchmarks while using 0.8‑4.9 GB VRAM (batch 1), a >5× savings over MolmoE‑7B and InternVL‑2B.

4. Comprehensive benchmark coverage OCR, charts, science QA, math, temporal reasoning, and movie QA； this breadth supports the generality claim.

5. This paper releases weights, training data mixtures, and code, plus a phone demo app, which is commendable for reproducibility and impact.

**Reasons To Reject:**

1.  VRAM figures are reported, but there are no wall‑clock throughput or energy measurements to substantiate “real‑time” claims, which are critical for edge deployment.

2. The paper notes spatial‑detail loss for OCR at high shuffle ratios yet adopts r = 4 for small models without a thorough error analysis on localisation‑heavy tasks.

3. Ablations missing for alternative compression. No comparison against patch‑merging, adaptive pooling, or transformer down‑sampling leaves open whether pixel shuffle is uniquely effective.

---

> ### Author Response · Authors · 2025-06-01
> **Response to R-ofar**
>
> We sincerely thank R-ofar for their positive evaluation, confidence in our work, and valuable feedback. We are pleased they recognize the significance of targeting edge devices and appreciate our systematic design-space exploration, comprehensive benchmarking, and commitment to open-source contributions, including the mobile application.
>
> Below, we address the points raised in their detailed review:
>
> 1. Real-time Claims: Wall-clock Throughput and Energy Metrics
>
> We agree that VRAM alone does not fully validate real-time performance. Thus, we will add additional figures with comprehensive wall-clock throughput metrics (Tokens Per Second, Time-To-First-Token) measured on both consumer hardware (MacBook M1-M4 Max series) and server-side GPUs (H100s). For example, SmolVLM-256M achieves up to 45 tokens/sec on a three-year-old MacBook Air M2 with 16GB of RAM, demonstrating practical real-time capabilities. *More information on this can be found in the general rebuttal.*
>
> Regarding energy efficiency, direct comparisons to other models across diverse hardware remain challenging; however, our results on the H100 GPUs indicate SmolVLM variants consistently outperform comparable models in tokens processed per second by at least 3 times, strongly suggesting superior energy efficiency per token.
>
>
> 2. Spatial Detail Loss with High Pixel Shuffle Ratios (r=4)
>
> We thank the reviewer for their valid concern regarding spatial detail loss with high shuffle ratios. Our adoption of r=4 for small models (reducing 512x512 patches from 1024 to 64 tokens) aimed to balance computational efficiency with general-purpose performance.
>
> Preliminary experiments at r=8 (16 tokens) showed significantly degraded performance after 24 hours on 64 H100 GPUs, making full training impractical and resource-prohibitive. Thus, r=4 was the most aggressive feasible compression. While a comprehensive error analysis on localization-heavy tasks for r=4 was not conducted, our reported OCR benchmarks are promising.
>
> In the revised manuscript, we will concisely discuss the trade-off of the r-parameter, our rationale for r=4, the limitations of our experiments (including the absence of the specific error analysis), and highlight detailed evaluation of shuffle ratios on localization-sensitive tasks as important future work.
>
>
>
> 3. Ablations for Alternative Compression Methods (e.g., Patch-merging, Adaptive Pooling)
>
> We acknowledge the reviewer's suggestion regarding ablations for alternative compression methods. We initially explored some options before deciding for pixel-shuffle as it is a straightforward yet highly effective compression method.
> Given our extensive existing analysis, exploring multiple additional compression techniques would significantly expand the scope. Tronchon et al. (Building and better understanding vision-language models: insights and future directions) explored some of these compression methods.
>
>
> In the revision, we will clearly state that comparative analysis of alternative compression methods is an exciting future research direction. Our current results provide a robust baseline for understanding their effectiveness.
>
>
>
> We appreciate R-ofar's insightful comments and strong support. We believe these clarifications and added analyses considerably strengthen our manuscript.

---

> > ### Comment · Reviewer_ofar · 2025-06-09
> >
> > After reading the rebuttal and comments from other reviewers, I adjust the score to 6 due to the feedback on the fairness of the experiment.

---

> ### Author Response · Authors · 2025-06-09
> **Response to R-ofar**
>
> Dear R-ofar,
>
> We conducted additional experiments to address the concerns raised by other reviewers regarding evaluation (measurement of TTPS and TPS, full table comparing more models in our response to R-RaaY).
>
> Do these not meet your concern, and if not, what additional comparisons/experiments would you need to see to address this concern?
>
> Best,
> The Authors

---

### Author Response · Authors · 2025-06-01
**Overall Response**

We sincerely appreciate the reviewers' insightful feedback, which will substantially enhance our manuscript. We are encouraged by the recognition of SmolVLM's *edge-efficient architecture*, particularly its ability to match or surpass significantly larger VLMs within a ≤1 GB VRAM constraint (R-ofar, R-RaaY, R-jgcB). Reviewers also commended our *systematic experimentation*, yielding clear design guidelines from an extensive ablation across encoder and LM scales, context length, shuffle ratios, and video sampling (R-ofar, R-RaaY, R-jgcB). Additionally, all reviewers highlighted the manuscript’s *full reproducibility*, noting the public availability of model weights, datasets, code, and a functional smartphone application as particularly valuable for immediate adoption and future research (R-ofar, R-RaaY, R-jgcB, R-BnL7). Finally, the broad evaluation spanning OCR, charts, science QA, mathematics, temporal reasoning, and movie QA was recognized as strong evidence of SmolVLM’s versatility and robustness (R-ofar, R-jgcB).

Across the reviews, we identified one primary recurring concern: the absence of real-time latency/throughput/energy metrics that fully substantiate our edge-efficiency claim. We have run additional experiments to validate this claim, detailed below.


## Enhanced Efficiency Metrics and Comparisons to SOTA Small VLMs (R-ofar, R-RaaY, R-jgcB)

We agree with R-ofar, R-RaaY, and R-jgcB that additional efficiency metrics beyond VRAM and direct comparisons with recent small VLMs are essential. To address this, we expanded our evaluation significantly:

### On-Device Performance
We benchmarked SmolVLM across various MacBook models (M1-M4 Max, 16GB-64GB RAM), achieving practical inference speeds of 20 tokens/sec (SmolVLM-2.2B, M1 Air) to 80 tokens/sec (SmolVLM-256M, M4 Max Pro) at batch size 1. We will add a figure to showcase this, and the raw results are:


*Table:* Generation TPS when running inference with a high-resolution [image](https://huggingface.co/spaces/merve/chameleon-7b/resolve/main/bee.jpg)
| Model Parameters | Pro M1 Max 64 GB | Pro M3 Pro 32 GB | Pro M2 Max 32 GB | Pro M4 Max 36 GB | Air M3 16 GB | Air M2 24 GB |
|------------------|-----------------|------------------|------------------|------------------|--------------|--------------|
| 256 M            | 39             | 66             | 57            | 78            | 52        | 45        |
| 500 M            | 40           | 47            | 50            | 55            | 32        | 36       |
| 2.2 B           | 33           | 26            | 34           | 44           | 13        | 12        |


*Real-Time Accessibility:* We will create public demos and will add links to them in the revised manuscript.
We used llama.cpp and transformers.js, alongside optimized ONNX exports for broader accessibility. These demos allow anyone to measure SmolVLM's real-time inference on their device.

### Server-Side Efficiency and Comparative Analysis:
H100 GPU Efficiency: Our revised manuscript will feature detailed H100 GPU benchmarks at practical input resolutions (up to 1536px), clearly demonstrating SmolVLM’s competitive performance:

*Table:* Generation TPS and TTFT comparison when running inference with a high-resolution image on an H100.
|  batch size = 1 | SmolVLM 256 | SmolVLM 500 | SmolVLM 2.2B | InternVL 2-2B | moondream2 | MolmoE-1B-0924 | MiniCPM-V-2 | Qwen2-VL-2B |
|----------------|-------------|-------------|--------------|---------------|------------|----------------|-------------|-------------|
| **TTFT (ms)**  | 36          | 45          | 71           | 180           | 88         | 492            | 306         | 435         |
| **Gen TPS**    | 40.4        | 35.8        | 39.7          | 37.4           | 55.6        | 5.3           | 24.4      | 17.9        |
|  **max batch size (MBS)** | 256 | 256 | 64 | 8 | 128 | 16 | 1* | 8 |
| **TTFT (ms) @MBS**  | 2890          | 2860          | 2920           | 2200          | 8560         | 3550            | 306*         | 6820         |
| **Gen TPS @MBS**    | 3509        | 2721        | 557          | 139           | 718        | 59.5           | 24.4*      | 45.5        |
*MiniCPM-V-2 official implementation only supports batch size=1 at inference time.



* Implication for Energy Efficiency: On a given hardware platform (e.g., H100), SmolVLM achieves significantly higher Tokens Per Second (TPS), showing superior energy efficiency, as more samples are processed within a similar power budget.

* Real-Time Suitability: Low TTFT (36-71ms at B=1) coupled with acceptable TPS (±40) highlights SmolVLM’s readiness for real-time deployment.


### Per-metric breakdown of the ablations
We will create a demo with a breakdown of all the experiments and the per-benchmark breakdown. Instead of plotting them in the manuscript which would reduce readability, readers will have access to interactive plots with these full evaluations.

---

### Decision · Program_Chairs · 2025-07-08

**Decision:**

Accept

**Comment:**

This paper introduces SmolVLM, a suite of compact vision-language models (256M–2.2B) optimized for edge devices under 1GB VRAM. The authors conduct extensive ablations and demonstrate strong performance across a wide range of image and video benchmarks. The work is well-motivated, reproducible (open-sourcing code, models, and a mobile app), and offers practical design insights.

Initial concerns about efficiency metrics and fairness of comparisons were addressed thoroughly in the rebuttal, including added throughput, latency, and broader baseline comparisons. While some techniques are incremental as pointed by the reviewers, the paper provides valuable engineering contributions with strong empirical backing. The Area Chairs believe this work will foster the development of cost-efficient and open-sourced VLM, and thus recommend an acceptance.